# Genome-wide transcriptomic analysis of the response of *Botrytis cinerea* to wuyiencin

**Liming Shi, Binghua Liu, Qiuhe Wei, Beibei Ge***, Kecheng Zhang* *

State Key Laboratory of Biology of Plant Diseases and Insect Pests, Institute of Plant Protection, Chinese Academy of Agricultural Sciences, Beijing, PR China

* zhangkecheng@sina.com (KZ); gbbcsx@126.com (BG)

**Data Availability Statement:** All relevant data are within the manuscript and its Supporting Information files. The RNA-seq dataset generated for this study can be found in the NCBI database under the accession number SRP212990.

## Abstract

Grey mould is caused by the ascomycetes *Botrytis cinerea* in a range of crop hosts. As a biological control agent, the nucleoside antibiotic wuyiencin has been industrially produced and widely used as an effective fungicide. To elucidate the effects of wuyiencin on the transcriptional regulation in *B. cinerea*, we, for the first time, report a genome-wide transcriptomic analysis of *B. cinerea* treated with wuyiencin. 2067 genes were differentially expressed, of them, 886 and 1181 genes were significantly upregulated and downregulated, respectively. Functional categorization indicated that transcript levels of genes involved in amino acid metabolism and those encoding putative secreted proteins were altered in response to wuyiencin treatment. Moreover, the expression of genes involved in protein synthesis and energy metabolism (oxidative phosphorylation) and of those encoding ATP-binding cassette transporters was markedly upregulated, whereas that of genes participating in DNA replication, cell cycle, and stress response was downregulated. Furthermore, wuyiencin resulted in mycelial malformation and negatively influenced cell growth rate and conidial yield in *B. cinerea*. Our results suggest that this nucleoside antibiotic regulates all aspects of cell growth and differentiation in *B. cinerea*. To summarize, some new candidate pathways and target genes that may related to the protective and antagonistic mechanisms in *B. cinerea* were identified underlying the action of biological control agents.

## Introduction

Grey mould is a type of disease that can be severe and economically damaging to many agricultural and horticultural crops [1]. *Botrytis cinerea* (teleomorph: *Botryotinia fuckeliana*) is an airborne plant pathogen with a necrotrophic lifestyle (class: *Leotiomycetes*, order: *Helotiales*, family: *Sclerotiniaceae*) [2]; it has been reported to cause serious losses in over 200 crop species worldwide. After winter dormancy, *B. cinerea* reportedly produces a large number of conidia under relatively suitable conditions in spring (the increase of relative humidity and temperature), which disperse via air and water droplets [3].

Airborne conidia usually cause new infections. Briefly, extracellular enzymes secreted by *B. cinerea*, such as pectin methylesterases (PMEs), polygalacturonases (PGs), laccases, and proteases, promote cell wall degradation and consequently soften host tissues. These changes

**Funding:** This work was supported by the National Natural Science Foundation (31601684), the Special Fund for Basic Scientific Research of the Chinese Academy of Agricultural Sciences (Y2017JC12), and the National Key Research and Development Plan (2017YFD0201100). The funders had no role in study design, data collection and analysis, decision to publish, or preparation of the manuscript.

**Competing interests:** The authors have declared that no competing interests exist.

consequently facilitate the penetration and colonization of host tissues, resulting in their decay [4, 5]. Some of the aforementioned enzymes express themselves differentially according to a variety of hosts and environmental factors, which explains how *B. cinerea* infects a broad range of hosts [6–8]. At times, *B. cinerea* is present on host plants in the latent state; this implies that while *B. cinerea* conidia do not adversely affect the host, when post-harvest fruits are stored and transported, they can germinate under conditions of relatively high humidity and suitable temperature, eventually causing serious damage [9].

Several chemicals have been widely used to tackle the problem of grey mould, but their prolonged usage has resulted in resistance development in *B. cinerea* and also given rise to strains that show rapid reproduction and genetic variations. More importantly, fungicide usage usually creates a problem of resistance, resurgence, and residue. Thus, to reduce environmental pollution, researchers have begun to screen and use beneficial microorganisms and their metabolites against *B. cinerea*. The nucleoside antibiotic wuyiencin is one such secondary metabolite; it is produced by *Streptomyces ahygroscopicus* var. wuyiensis, which was first isolated from the natural soil habitat of Wuyi Mountain in China [10]. After being industrially produced (COFCC-R-0903-0070), wuyiencin has been extensively used to control various fungal diseases in vegetables and crops and to enhance their resistance to diverse pathogens. It can be regarded as an organic, pollution-free pesticide, considering its characteristics of high efficiency, broad spectrum, and low toxicity [10]. Wuyiencin can alter cytomembrane permeability and inhibit protein synthesis in the mycelium of *B. cinerea*, which consequently causes cytoplasm leakage and mycelial malformation, thus reducing the pathogenicity of *B. cinerea*. It also reportedly induces host plant resistance to pathogenic bacteria [11]. However, little is known about the molecular mechanisms underlying the action of wuyiencin, particularly in *B. cinerea*.

In the past few years, many researchers have focused on studying and controlling *B. cinerea*; it has in fact become one of the important model systems in molecular phytopathology. The first genome assemblies of two *B. cinerea* strains, B05.10 and T4, were sequenced using Sanger technology at low coverage [12, 13]. And the gapless, near-finished genome sequence of *B. cinerea* was reported in 2017 [14]. Genome sequences of *B. cinerea* have played a major role in facilitating genetic manipulations and analyzing the genetic basis of pathogenicity [15]. Moreover, high-coverage *de novo* assemblies of genome sequences have promoted the development of genome-wide transcriptomic and proteomic techniques in *B. cinerea* [5]. RNA sequencing (RNA-seq) and transcriptomic analyses are commonly used methods as they are very sensitive, quantitative, accurate, and affordable [16].

Here we performed a genome-wide transcriptomic analysis to study the response of *B. cinerea* to wuyiencin. According to our results, wuyiencin had a prominent effect on the expression of genes involved in, for example, amino acid metabolism, protein synthesis, DNA replication, and cell cycle. Moreover, it caused mycelial malformation and negatively influenced cell growth rate and conidial yield in *B. cinerea*.

## Results

### Influence of wuyiencin on the growth and morphology of *B. cinerea*

In response to varying concentrations of wuyiencin, *B. cinerea* growth was gradually inhibited; aerial mycelia and pigment production were reduced as well (Fig 1A). Notably, with an increase in wuyiencin concentration, the antibiotic resulted in tortuous, malformed mycelia, the branching decreased, and the hyphal tip expanded to form spherical vesicles (Fig 1B). In response to 50 μg/mL, 100 μg/mL, and 200 μg/mL wuyiencin, the cell growth rate of *B. cinerea* decreased by 25.58%, 43.95%, and 100.00%, respectively, and conidial yield declined by

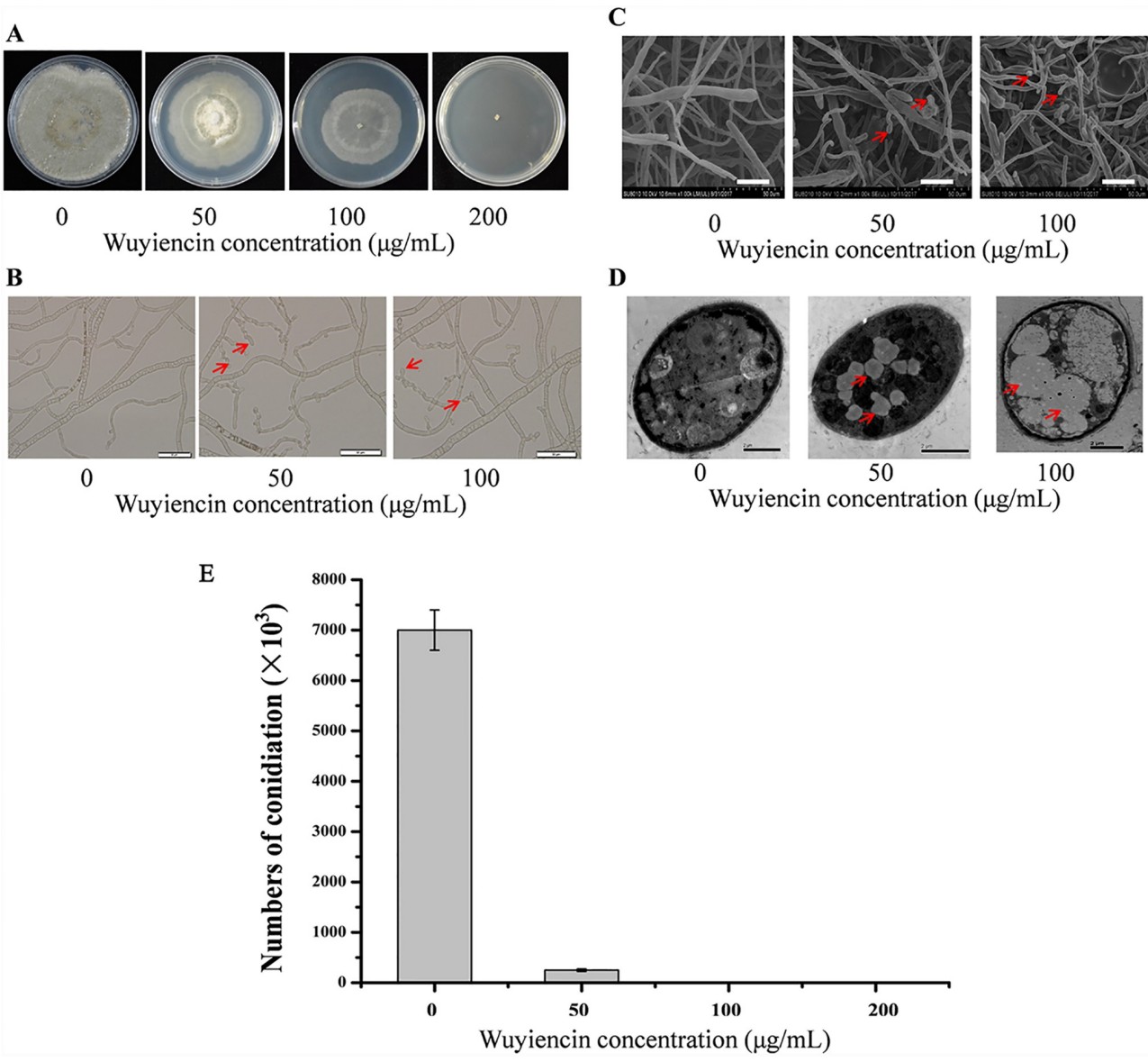

**Fig 1. Phenotypes of *Botrytis cinerea* cultivated with various concentrations of wuyiencin (50 μg/mL, 100 μg/mL, and 200 μg/mL wuyiencin). (A)** *B. cinerea* colonies morphology; **(B)** mycelial morphology, as observed under a light microscope, scale bars: 50 μm; **(C)** mycelial morphology, as observed using transmission electron microscopy (TEM), scale bars: 25 μm; **(D)** subcellular structure, as observed using TEM, scale bars: 2 μm; and **(E)** conidial production in *Botrytis cinerea* in response to varying concentrations of wuyiencin.

96.43%, 99.90%, and 100.00%, respectively (Fig 1E). Mycelial morphology and subcellular structure of *B. cinerea* were observed using SEM and TEM (Fig 1C and 1D). In comparison with the control treatment, when *B. cinerea* was cultivated with wuyiencin, we noted not only mycelial abnormality and severe hyphal swelling but also vesicular fusion; moreover, the number of organelles in mycelium decreased and autophagic bubbles with double membrane appeared (Fig 1C and 1D). These results indicated that wuyiencin could significantly inhibit the cell growth rate and conidial yield in *B. cinerea*, causing mycelial malformation. As cell growth and conidiogenesis in *B. cinerea* are related to its pathogenicity, our data suggest that wuyiencin can considerably weaken the pathogenicity of *B. cinerea*.

## Genome-wide transcriptomic analysis

We performed a genome-wide transcriptomic analysis to elucidate the molecular mechanisms underlying the action of wuyiencin. The final concentration of wuyiencin is approximately 50–100 μg/mL in agricultural production; earlier studies [11] have reported that 200 μg/mL wuyiencin is lethal to *B. cinerea*, but there is little effect on cell growth and conidiogenesis at 50 μg/mL wuyiencin. Thus, for RNA-seq analyses, we cultivated B05.10 in the dark at 20˚C with 100 μg/mL and without wuyiencin. Three biological replicates were assessed for each sample.

Of the 12814 genes in *B. cinerea*, 2067 genes were differentially expressed in response to treatment with 100 μg/mL wuyiencin (Fig 2); of them, 886 and 1181 were significantly upregulated and downregulated, respectively. Gene ontology (GO) and Kyoto encyclopedia of genes and genomes (KEGG) enrichment analysis were performed to classify all DEGs into the following functional categories: amino acid metabolism, protein synthesis, carbon and energy metabolism, putative secreted metabolites and proteins, DNA replication and cell cycle, and other metabolism.

## Transcriptional analysis of genes involved in amino acid metabolism and protein synthesis

Amino acid metabolism and protein synthesis are fundamental cellular activities. The expression of most genes involved in the metabolism of aromatic amino acids, alanine, glycine, serine, threonine, cysteine, and methionine was upregulated when *B. cinerea* was cultivated with wuyiencin (S1 Table).

A major change in the metabolism of tyrosine, phenylalanine, and tryptophan was observed. These aromatic amino acids play a significant role in protein synthesis, and they also participate in the synthesis of various secondary metabolites. The structural and catalytic properties of these amino acids give specific functions to certain proteins [17]. In fungi, the terminal reactions in the biosynthesis of phenylalanine and tyrosine are achieved by converting chorismite to phenylpyruvate or 4-hydroxyphenylpyruvate, and tyrosine aminotransferase (dependent on pyridoxal-5-phosphate) catalyzes the final step in both pathways with glutamate as an amino donor [18]. Owing to the promiscuous substrate specificity of aminotransferases in the anabolism of aromatic amino acids, branched-chain and aspartate aminotransferases have activities that overlap with those of tyrosine aminotransferase [19–21]. Noticeably, having different aminotransferases with overlapping substrates would be a strategy to achieve nutritional flexibility in evolution under various growth conditions [22]. In *B. cinerea*, the transcriptional levels of the following key enzymes were significantly upregulated upon wuyiencin treatment: tyrosine aminotransferase (Bcin07g04780), branched-chain aminotransferase (Bcin04g01520), and L-ornithine aminotransferase (Bcin07g00730). This indicates that *B. cinerea* responds to wuyiencin by increasing tyrosine, phenylalanine, and tryptophan biosynthesis.

However, the expression of some key enzymes was significantly downregulated. Bcin15g05090 and Bcin16g01460 expression was downregulated ($Log_2FC$ values, −2.0 and −4.4, respectively); both are involved in the conversion of tyrosine to L-3,4-dihydroxyphenylalanine (L-DOPA), a reaction catalyzed by tyrosine hydroxylase, according to KEGG enrichment analysis. The ubiquitous enzyme aromatic L-amino acid decarboxylase (Bcin06g01710) is involved in the decarboxylation of L-DOPA to dopamine, of 5-hydroxytryptophan to serotonin, and of tryptophan to tryptamine [23]. Furthermore, amine oxidases (Bcin01g08900, Bcin05g07030) convert dopamine to 3,4-dihydroxyphenylacetaldehyde and also phenylethylamine to phenylacetaldehyde. As per our results, Bcin06g01710, Bcin01g08900, and

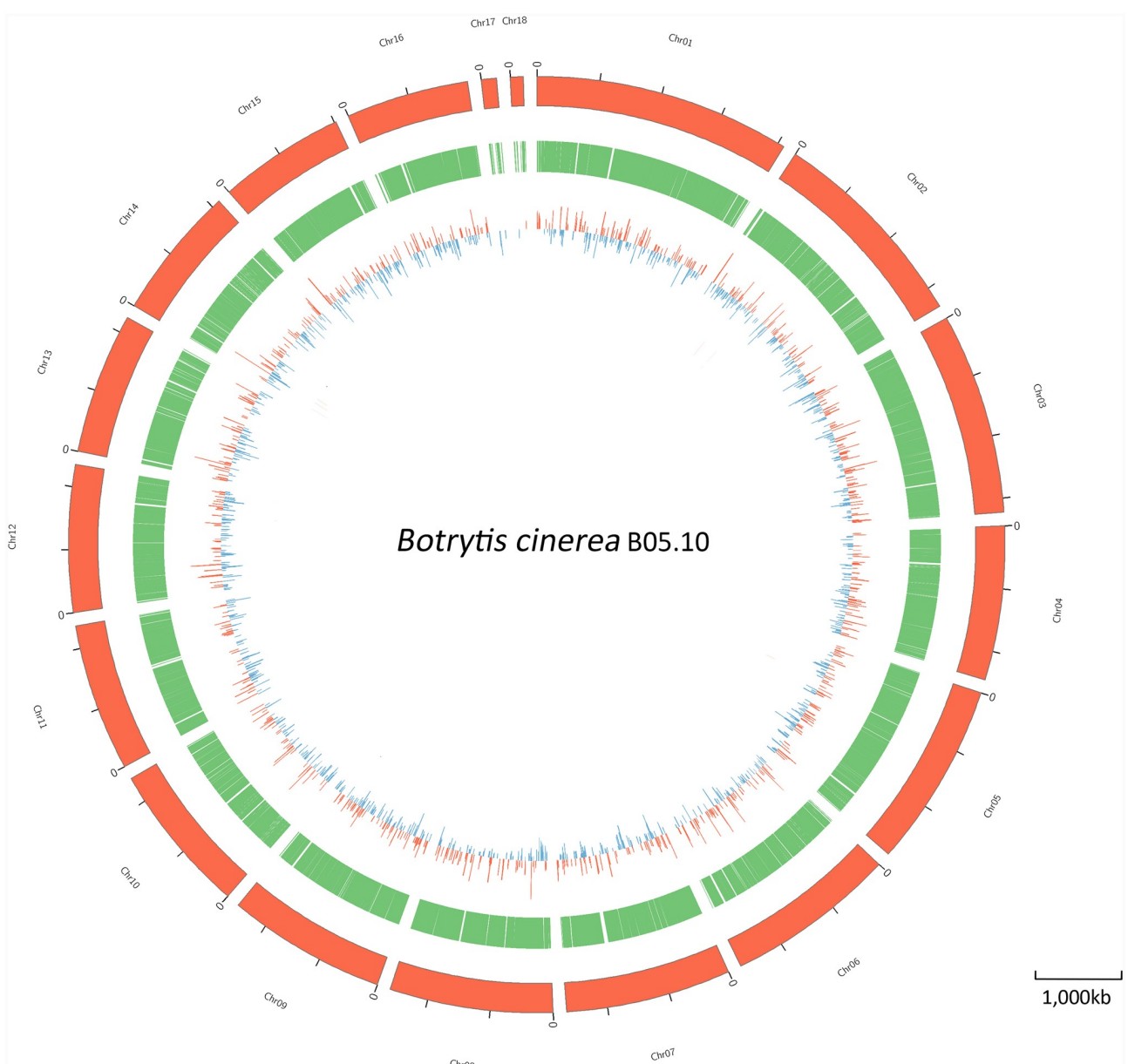

**Fig 2. Circular map of B05.10 genome and genes that were differentially expressed (DEGs) upon wuyiencin treatment.** Moving inward, the outer two rings show the length (red ring) and gene (green ring) density of every chromosome, respectively. The third ring represents differentially expressed genes (DEGs): upregulated and downregulated DEGs are highlighted as red and blue bars, respectively. The length of each bar represent the fold change.

Bcin05g07030 expression was significantly downregulated (Log$_2$FC values, −2.9, −3.2, and −5.8, respectively). Although the expression of Bcin15g03370, which encodes 4-hydroxyphenylpyruvate dioxygenase that transforms 4-hydroxyphenylpyruvate to homogentisate, was slightly upregulated, that of two genes (Bcin11g01020 Bcin11g01050) involved in the conversion of homogentisate to fumarate and acetoacetate in tyrosine metabolism and another one (Bcin11g01060) encoding phenylacetate 2-hydroxylase, which hydroxylates phenylacetate to 2-hydroxy phenylacetate for styrene degradation, was markedly downregulated. The downregulation of these genes was in stark contrast to the upregulation of those involved in aromatic

amino acid synthesis. These findings suggest that wuyiencin has an effect on both aromatic amino acid anabolism as well as catabolism.

Moving on, the expression of three key enzymes involved in valine, leucine, and isoleucine degradation was significantly upregulated (Fig 3A). After the transamination of branched-chain aminotransferase (Bcin04g01520), valine, leucine, and isoleucine are degraded from 2-oxobutanoate or 2-oxopentanoate to butyryl-CoA or propionyl-CoA by 3-methyl-2-oxobu-tanoate dehydrogenase (Bcin13g01430 and Bcin13g03020) and dihydrolipoamide acetyltrans-ferase (Bcin11g04250). The $Log_2FC$ value of Bcin13g01430, Bcin13g03020 and Bcin11g04250 were 2.6, 2.0, and 3.4, respectively (Fig 3B, S1 Table). Concomitantly, the transcriptional levels of enoyl-CoA hydratase (Bcin03g05840) and acetyl-CoA acyltransferase (Bcin01g04960), which convert butanoyl-CoA to propanoyl-CoA, were also significantly upregulated. Similarly, the expression of genes involved in the transamination of cysteine metabolism (Bcin04g01520, Bcin14g01940) and glycine, serine, and threonine metabolism (Bcin04g03090) was upregu-lated, leading to an increase in 2-oxobutanoate and pyruvate production. To supportour RNA-seq data, we randomly chose Bcin07g04780, Bcin01g08900, Bcin05g07030, Bcin15g03370, and

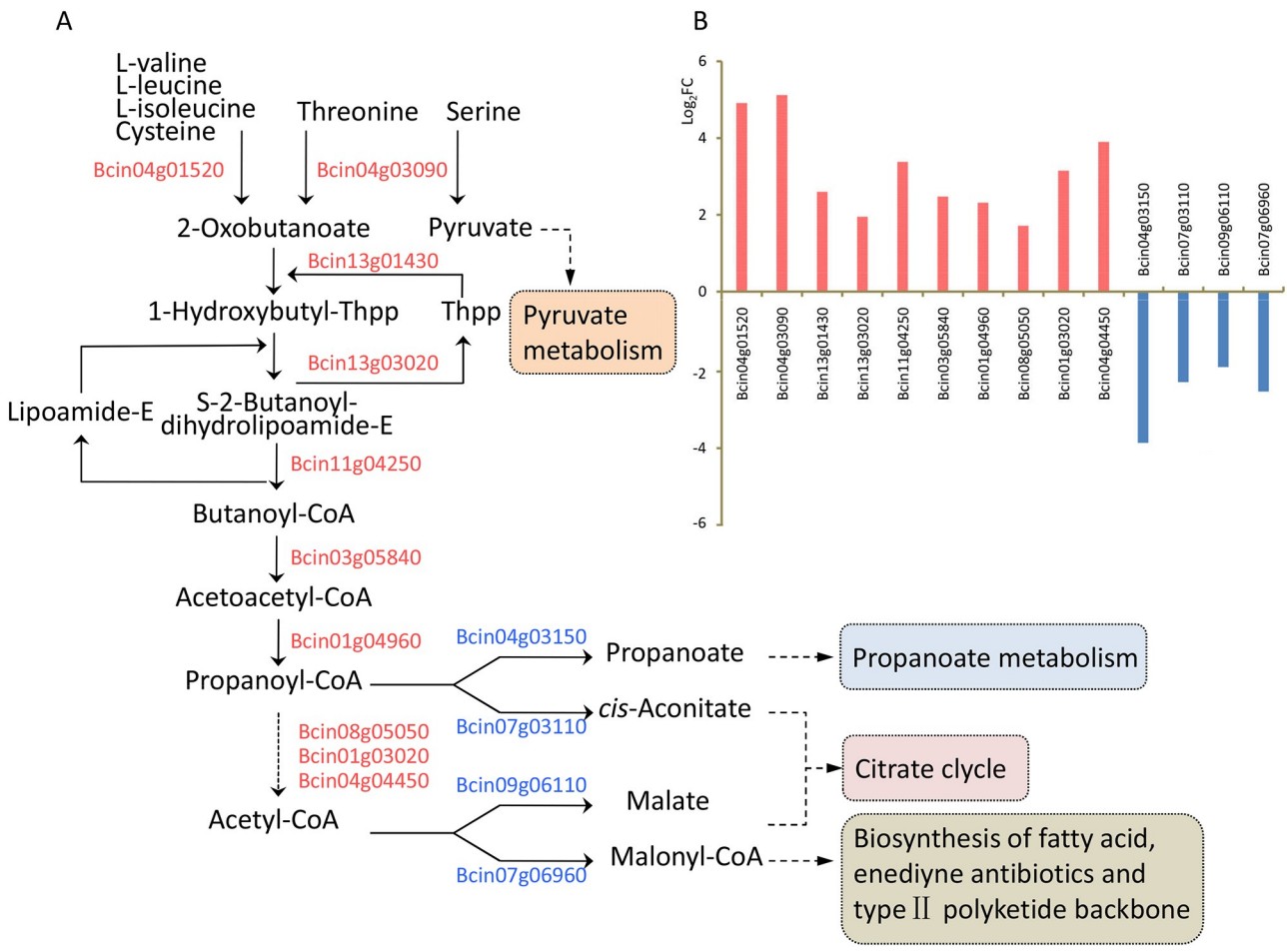

**Fig 3. Expression profiles of genes involved in amino acid metabolism. (A)** Schematic representation of differentially expressed genes (DEGs) involved in amino acid metabolism in *Botrytis cinerea*; red and blue gene IDs indicate upregulated and downregulated genes, respectively; **(B)** $Log_2FC$ profile of DEGs involved in amino acid metabolism. FC represents the fold change in read count in response to cultivating *B. cinerea* with 100 μg/mL and without wuyiencin.

Bcin11g04250 for RT-qPCR analyses, and RT-qPCR findings supported our RNA-seq data (S2 Table). These data indicate that the substrate levels for carbon metabolism (examples being propanoate metabolism and citrate cycle) may increase and that wuyiencin promotes the catabolism of some amino acids.

Among the genes associated with protein synthesis in *B. cinerea*, there were 28 genes enriched in term "ribosome" by KEGG, and their expression was found to be significantly upregulated (S1 Table). Most of them encoded ribosomal proteins, including 40S ribosomal proteins S5 (Bcin03g00590), S14 (Bcin03g06970), S23 (Bcin04g05440), and S25 (Bcin03g00790), 60S ribosomal proteins L10ae (Bcin03g01820), L15 (Bcin14g04790), L24 (Bcin09g06480), L26 (Bcin13g03600), L27ae (Bcin02g05140), L29 (Bcin10g03580), L30 (Bcin09g05600), L37 (Bcin01g00770), L38 (Bcin05g06070), L43 (Bcin13g00830), and L44 (Bcin09g02490), and 60S ribosome biogenesis protein SQT1 (Bcin08g05590). A previous gene-profiling study on *B. cinerea* germination reported that transcripts encoding ribosomal proteins were markedly affected upon resveratrol treatment [24]. The expression of the 40S ribosomal protein S17 (Bcin12g01300), 60S ribosomal proteins L18ae (Bcin05g01600) and L9 (BC1G_16427), and 60S ribosome biogenesis protein SQT1 (Bcin08g05590) was significantly downregulated, but that of 40S ribosomal proteins S8 (Bcin04g05190) and S5 (Bcin03g00590) and 60S ribosomal proteins L14 (Bcin03g06970), L44 (Bcin09g02490), L9 (Bcin11g05920), and L35 (Bcin11g03280) was significantly upregulated [24]. However, only the trend shown by the 60S ribosomal protein L44 (Bcin09g02490) was the same when *B. cinerea* was treated with wuyiencin or resveratrol; all other genes were either not differentially expressed or showed contradictory expression trends upon wuyiencin treatment. These findings suggest that *B. cinerea* uses different protein synthesis metabolism when exposed to phytoalexin from plants or antibiotic from microorganisms. In addition, we identified only one gene encoding a chaperone (Bcin01g09530) that was involved in protein folding, but the expression of this heat-shock protein (HSP), belonging to the HSP 20 family, with the highest $Log_2FC$ value (4.7) among the genes in protein synthesis metabolism.

## Transcriptional analysis of genes involved in carbon and energy metabolism

With regard to propanoate metabolism, the expression of genes encoding glucan 1,4-alpha-glucosidase (Bcgs1, Bcin15g05660), 2-methylcitrate dehydratase (Bcin07g03110), and propionate-CoA ligase (Bcin04g03150) was markedly downregulated (Fig 3A, S1 Table). Remarkably, the necrosis-inducing glycoprotein BcGs1 (Bcin15g05660) produced by *B. cinerea* has elicitor activity and triggers defense response in host plants [25]. With regard to pyruvate metabolism, the expression of genes encoding L-lactate dehydrogenase (Bccyb2, Bcin01g00400), acetyl-CoA carboxylase (Bcin07g06960), pyruvate carboxykinase (Bcpck1, Bcin16g00630), and malate synthase (Bcin09g06110) was also significantly downregulated (Fig 3A). Though this downregulation might lead to a decrease in the production of pyruvate, malate, and malonyl-CoA, the genes involved in glycolysis, gluconeogenesis, and citrate cycle were not differentially expressed in response to wuyiencin treatment.

With respect to energy metabolism, six genes involved in oxidative phosphorylation were differentially expressed. Two of them, Bcin01g01100 encoding NADH dehydrogenase in complex 1 and Bcin04g00750 encoding plasma membrane ATPase in complex 5 (ATP synthase), were considerably downregulated, whereas the other four, encoding NADH oxidoreductase (Bcin10g04670 and Bcin14g00100), ATP synthase protein (Bcin10g01500), and cytochrome C oxidase (Bcin02g03350), were significantly upregulated. Moreover, the expression of Bcfdh1 (Bcin16g04640), encoding formate dehydrogenase which is potentially implicated in nitrate respiration in *B. cinerea* and supplies electrons, was upregulated ($Log_2FC$ value, 2.6). This

finding was, however, contrary to that reported by Zheng et al. [24]; as per their results, Bcfdh1 expression was downregulated on cultivating *B. cinerea* with resveratrol. Further, the expression of Bcin03g01010, which encodes a ADP/ATP mitochondrial carrier protein, was downregulated at the transcriptional level. This carrier protein is a mitochondrial translocase and promotes the transport of solutes across the mitochondrial membrane; this process between the mitochondrion and cytosol is dependent on ATP [26]. This finding was also contrary to the results reported by Zheng et al. [24]. It is noteworthy that RT-qPCR results for Bcin01g00400, Bcin07g06960, Bcin02g03350, and Bcin16g04640 were all basically consistent with our RNA-seq data (S2 Table).

## Transcriptional analysis of genes encoding putative secreted metabolites and proteins

As a natural barrier, the plant cell wall provides mechanical strength and rigidity in order to prevent invasion by pathogens. To destroy the plant cell wall and successfully colonize host plants, *B. cinerea* secretes a diverse array of metabolites and proteins [25, 27, 28]. These proteins belong to cell wall-degrading enzyme (CWDE) families and include pectinases, xylanases, and endo-PGs; they are usually considered to be important virulence factors and act via host tissue impregnation and macromolecule degradation [29]. In response to cultivating *B. cinerea* with wuyiencin, the expression of 23 CWDE family-related genes was considerably influenced (S1 Table).

To degrade xylan, the major hemicellulosic component of the plant cell wall, *B. cinerea* needs the synergistic action of several hydrolytic enzymes. One such enzyme is endo-β-1,4-xylanase, which is encoded by the gene *Bcxyn11A* (Bcin03g00480). This enzyme belongs to the glycosyl hydrolase family 11 and carries out the initial breakdown of the xylan backbone [30]. For pectin degradation, *B. cinerea* employs a series of depolymerizing enzymes. PME is involved in one such pathway, which starts with pectin de-esterification into methanol and polygalacturonic acid (PGA). There are two genes, *Bcpme1* (Bcin08g02970) and *Bcpme2* (Bcin03g03830) (Fig 4A and 4B). PME activity is also important for the subsequent action of depolymerizing enzymes, which, depending on their mode of action, are classified into two groups: those of endo cleavage mode, which is random, and those of endo cleavage mode, which act on the penultimate polymer bonds [31]. Then, glycosidic bonds are interrupted by PG hydrolysis (encoded by *Bcpg1*, Bcin14g00850 and endo-PG; *Bcpg2*, Bcin14g00610; and *Bcpg5*, Bcin01g07330) and PGA is broken down into oligogalacturonides by the β-elimination of pectate lyases (endo-PL, Bcin03g05820). In plant–pathogen interactions, CWDEs act not only as triggers of pathogen-associated molecular pattern-triggered immunity responses in plants but also as virulence factors. In particular, in *B. cinerea*, xylanase (encoded by *Bcxyn11A*, Bcin03g00480) and endo-PG 1 (encoded by *Bcpg1*, Bcin14g00850) trigger immune responses and play a vital role in virulence [32]. The expression of these seven genes (Bcin03g00480, Bcin08g02970, Bcin03g03830, Bcin14g00850, Bcin14g00610, Bcin01g07330 and Bcin03g05820) was noted to be significantly upregulated (Fig 4B); the RNA-seq data generated for Bcin08g02970 and Bcin03g00480 were supported by RT-qPCR, as standard (S2 Table).

In contrast, several genes encoding putative secreted carbohydrate-active enzymes (CAZymes) were differentially expressed (Fig 4C). CAZymes are proteins with certain catalytic and carbohydrate-binding domains that degrade, modify, or create glycosidic bonds [33]. The expression of genes encoding xyloglucan (XyG) backbone-degrading enzymes was significantly upregulated. The product (glycoside hydrolase family 16 protein, GH16) of Bcin01g06010 belongs to XyG hydrolases, and the XyG backbone is hydrolyzed by endo-acting β-1,4-glucanases (encoded by Bcin03g03630) [34]. The $Log_2FC$ values for Bcin01g06010 and

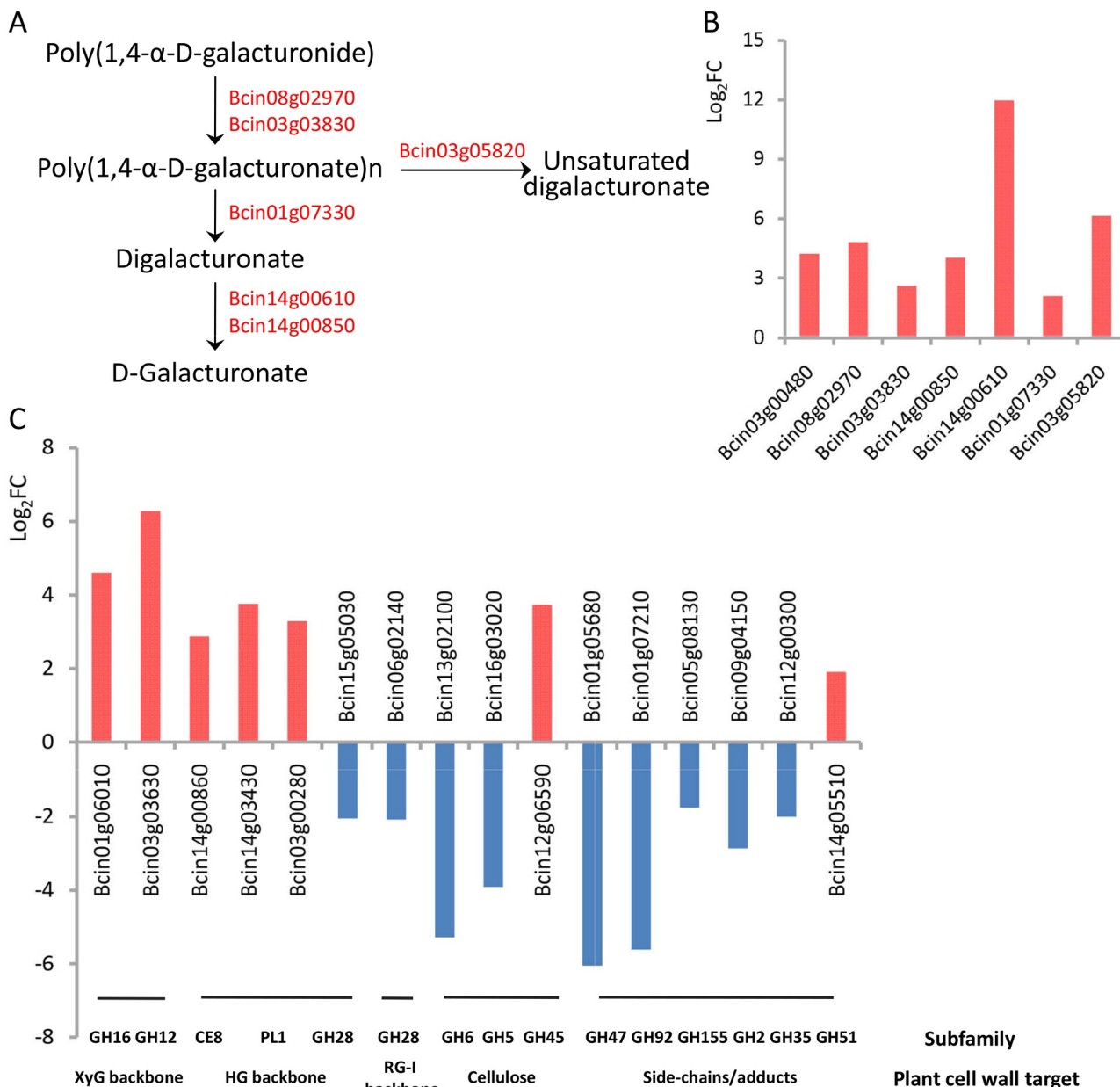

**Fig 4. Expression profiles of genes encoding putative secreted metabolites and proteins.** (**A**) Schematic representation of differentially expressed genes (DEGs) encoding cell wall-degrading enzymes (CWDEs) in *Botrytis cinerea*; red gene IDs indicate upregulated genes; (**B**) Log$_2$FC profile of DEGs encoding CWDEs; (**C**) Log$_2$FC profile of DEGs encoding putative secreted carbohydrate-active enzymes.

Bcin03g03630 were 4.6 and 6.3, respectively, which were among the highest recorded in this study. In the primary cell wall of dicots and non-graminaceous monocots, the major pectic polysaccharides are homogalacturonan (HG) and rhamnogalacturonan (RG) [35]. We noted that the expression of almost all CAZymes involved in cleaving HG pectin backbones was upregulated at the transcriptional level, including pectinesterase (Bcin14g00860) and pectin lyase (Bcin14g03430 and Bcin03g00280). In fact, PG (*Bcpg1*, *Bcpg2*, and *Bcpg5*) and PME (*Bcpme1* and *Bcpme2*) also belong to this class. However, glycoside hydrolase family 28 proteins (GH28) to the RG-I backbone (Bcin06g02140) and HG backbone (Bcin15g05030)

showed low expression levels. Notably, among the genes encoding CAZymes targeting cellulose, side-chains/adducts, and xylan backbone, the expression of nearly all genes was downregulated, except the expression of Bcin12g06590 and Bcin14g05510, which was slightly upregulated. Our data thus indicate that the expression of these genes (Bcin15g05030, Bcin06g02140, Bcin13g02100, Bcin16g03020, Bcin01g05680, Bcin01g07210, Bcin05g08130, Bcin09g04150 and Bcin12g00300) is suppressed by wuyiencin.

## Transcriptional analysis of genes involved in DNA replication and cell cycle

The expression of 12 genes related to DNA replication and cell cycle was downregulated when *B. cinerea* was treated with wuyiencin. With regard to DNA replication, the expression of *Bcpol1* (Bcin16g02310), *Bcpol12* (Bcin08g00800), and *Bcpri2* (Bcin08g00460), encoding the components of DNA polymerase α-primase complex, was downregulated, with values of −2.6, −2.0, and −1.9 Log$_2$FC, respectively (S1 Table). *Bcpol1* encodes Bcin16g02310, which also responds to DNA damage stimulus and cell differentiation. Besides, transcripts for Bcin08g05510 and Bcin02g07590, which encode the components of DNA polymerase δ- and ε-complex, respectively, were downregulated (Log$_2$FC values, −1.9 and −1.8, respectively). The expression of genes encoding DNA helicase (Bcin15g02990 and Bcin12g06740), proliferating cell nuclear antigen (Bcin01g06320), and replication protein A (Bcin12g01760) were also significantly downregulated. The expression of three cell cycle-related genes was also downregulated: *Bccdc6* (Bcin01g06090, encoding cell division control protein), *Bcmet30* (Bcin07g06910, encoding E3 ubiquitin ligase complex SCF subunit), and *Bcmrc1* (Bcin09g01360, participating in cell cycle arrest). The downregulation of the expression of these genes may be related to cell growth impairment in *B. cinerea* upon wuyiencin treatment. To assess if RNA-seq data was supportable by a different method of analysis, several genes (Bcin16g02310, Bcin08g00460, Bcin01g06320 and Bcin07g06910) were randomly selected for RT-qPCR, and the results were basically consistent with RNA-seq data (S2 Table).

## Transcriptional analysis of genes involved in other metabolism

With regard to vacuole membrane metabolism, some genes that probably participate in fungal physiology and morphogenesis were differentially expressed in response to wuyiencin (S1 Table). Their differential expression might be related to changes in cell morphogenesis, characterized by vesicular fusion and severe hyphal swelling (Fig 1). Three genes were enriched in fungal-type vacuole membrane and lytic vacuole membrane metabolism, as elucidated via GO analysis. The expression of Bcin08g03620 (encoding an uncharacterized protein), related to amide and peptide transporter activity and ATPase activity coupled to the movement of substances, was upregulated, whereas that of Bcin03g05090 (also encoding an uncharacterized protein), related to inorganic cation transmembrane transporter activity, and Bcin12g01770, encoding acetyltransferase, was significantly downregulated (S1 Table). Moreover, with regard to fungal cell wall metabolism, the expression of *Bccrh1* (Bcin01g06010), encoding GPI-glucanosyltransferase from the glycosyl hydrolase family 16 (GH16), was considerably upregulated (S1 Table). Reportedly, this GPI-glucanosyltransferase is a homolog of the CRH1 protein that is necessary for the cross-linking of chitin to β1,6-glucan in *Saccharomyces cerevisiae*; *Bccrh1* expression was significantly upregulated when *B. cinerea* was treated with resveratrol too [24, 36]. Thus, it seems like the product of *Bccrh1* directly participates in the formation of cross-links between cell wall components [11].

As to fungal transporters in *B. cinerea*, the expression of four genes encoding ATP-binding cassette (ABC) or major facilitator (MFS) transporters was significantly upregulated (S1

Table). It is known that fungi employ active efflux by ABC and MFS transporters to resist endogenous and exogenous toxic compounds, such as antibiotics and fungicides [37–39]. In *B. cinerea*, the expression of *BcatrB* (Bcin13g00710, encoding an ABC transporter) and *Bchex1* (Bcin09g00150, encoding an MFS sugar transporter) was markedly upregulated, with Log$_2$FC values reaching as high as 6.2 and 1.8, respectively. These results suggest that the active efflux of wuyiencin occurs in *B. cinerea*. This observation is also consistent with the finding that the expression of ABC and MFS transporters is significantly upregulated on cultivating *B. cinerea* with resveratrol [24]. Besides, the expression of MFS transporter genes *Bcmfs1* (Bcin14g02870) and *BcmfsM2* (Bcin15g00270) was markedly upregulated too; they are also involved in multi-drug resistance, such as in transporting azole fungicides [40]. Moreover, ABC transporters contribute to *B. cinerea* virulence, and BcATRB is a known virulence factor [39]. These data indicate that the upregulation of the expression of these genes leads to an increase in virulence. As standard, the RNA-seq data generated for Bcin08g03620 and Bcin13g00710 were supported by RT-qPCR (S2 Table).

## Discussion

Although wuyiencin has been industrially produced and widely used in China as an effective fungicide, the biological mechanisms by which this antibiotic inhibits pathogenic fungi still remain unclear [41]. We, for the first time, performed a genome-wide transcriptomic analysis to study the response of *B. cinerea* to wuyiencin treatment. In an earlier study to evaluate the *Botrytis*–wuyiencin relationship, including morphological characterization, it was reported that wuyiencin significantly reduces the production of mycelial proteins and almost completely inhibits the germination of conidiospores [11]. In this study, mycelial morphology and cell growth rate were intensely influenced in a concentration-dependent manner and conidial germination was completely inhibited upon exposure to 100 μg/mL wuyiencin; it is noteworthy that Sun et al. [11] also used a similar concentration of wuyiencin.

At present, with the development and utilization of fungicides for agricultural use, the mechanisms by which they act at a morphological, physiological, and molecular level to inhibit the growth of pathogenic microorganisms are gradually been elucidated. For example, nucleoside antibiotics are a diverse group of microbial secondary metabolites derived from nucleosides and nucleotides, which play several roles in basic cellular metabolic pathways [42]. Polyoxin and nikkomycin are typical antifungal nucleoside antibiotics that target cell wall biosynthesis and serve as competitive inhibitors of fungal chitin synthases [43]. In the fungal cell wall, the rigid carbohydrate polymer chitin is a significant structural component that provides strength and rigidity. Thus, the inhibition to chitin biosynthesis by antibiotics can cause severe hyphal swelling, consequently affecting fungal growth [44]. Moreover, the aminoglycoside antibiotic kasugamycin is known to inhibit protein biosynthesis and is widely used for managing plant diseases [45]. Kasugamycin inhibits the binding of ribosomes and aminoacyl-tRNA-message complex, but it does not affect nucleic acid synthesis [46]. Natamycin and other polyene antibiotics, which are widely used as antifungal agents, act by destroying the barrier function of cell membrane [47]. Polyene antibiotics usually consist of a macrolide core with three to eight conjugated double bonds, an exocyclic carboxyl group, and an unusual mycosamine sugar [48]. Natamycin acts by binding to principal sterols in fungal membranes, specifically to ergosterol [47]. With regard to the nucleoside antibiotic wuyiencin, not only little is known about its underlying molecular mechanism of action in *B. cinerea* but also the transcriptomic changes in *Botrytis* species upon wuyiencin treatment have not been previously studied.

In the present study, 2067 genes were differentially expressed in B05.10 after wuyiencin treatment; of these, 886 were upregulated and 1181 were downregulated. This suggests that wuyiencin activates the expression of some genes and stimulates a resistance mechanism in *B. cinerea*, while inhibiting the expression of some other genes. Further, wuyiencin had a remarkable effect on primary metabolism in *B. cinerea*, and many genes involved in amino acid metabolism were enriched. Pertaining to the metabolism of three aromatic amino acids, the expression of genes involved in anabolism was upregulated, while that of genes participating in catabolism was downregulated. After transamination and hydrolysis, aromatic amino acids are transformed to fumarate and acetoacetate, thereby linking the metabolism of amino acids and fumarate, potentially both inside and outside of mitochondria [17]. We noted that the expression of some genes encoding key enzymes involved in valine, leucine, and isoleucine degradation was upregulated; with regard to carbon metabolism, some genes encoding key enzymes showed low expression levels. This could decrease the substrate levels for pyruvate metabolism, citrate cycle, and fatty acid biosynthesis. In addition, genes related to protein synthesis and most genes involved in energy metabolism (oxidative phosphorylation) showed upregulated expression levels, which is contradictory to the response of *B. cinerea* to resveratrol treatment [24]. In particular, the expression of genes (Bcin01g06320, Bcin07g06910, Bcin08g03620, Bcin16g02310) involved in stress response was significantly downregulated; the products of these genes are involved in responding to DNA damage stimulus, metal ions, and toxic substances. Also, the expression of all genes involved in DNA replication and cell cycle and which were directly related to cell growth was suppressed by wuyiencin. The expression of genes encoding CAZymes was also significantly impacted upon wuyiencin treatment, indicating that gene expression might either be influenced by wuyiencin or *B. cinerea* adjusted the virulence and host tissue targets proactively.

Our results are in agreement with those reported by a previous study and suggest that wuyiencin plays an important role in regulating of all aspects of cell growth and differentiation in *B. cinerea*, in view of the broad spectrum of biological activities of nucleoside antibiotics [43]. From the experimental data, the changes caused by wuyiencin were partly similar with general stress response and induced ROS reaction [49, 50]. The relationship between biotic and abiotic stress of *B. cinerea* to wuyiencin need more investigation to clarification. We hypothesized that the transcriptomic response of *B. cinerea* to wuyiencin involves striking a balance between antagonism and competition. More importantly, in this study, some new candidate pathways and target genes that may related to the protective and antagonistic mechanisms in *B. cinerea* were identified underlying the action of biological control agents.

In conclusion, to further validate our results, transcriptomic and proteomic analyses in actual field cultivation situations should be performed. The signal transduction pathway and regulatory mechanisms of wuyiencin that affect the growth and development of *B. cinerea* need to be elucidated in future studies.

## Materials and methods

### Treatment of *B. cinerea* with wuyiencin

The *B. cinerea* standard strain B05.10 was separately inoculated into potato dextrose agar (PDA) medium containing no wuyiencin (control) and that containing wuyiencin at a final concentration of 50 μg/mL, 100 μg/mL and 200 μg/mL. Cultures were incubated at 20˚C, and mycelial morphology was observed under a light microscope after 3 days. Conidia were eluted with 0.2% Tween 20 after 7 days and quantified using a hemocytometer under a microscope.

## Scanning Electron Microscopy (SEM) and Transmission Electron Microscopy (TEM)

B05.10 was inoculated into PDA medium with and without wuyiencin, as mentioned above. Cultures were incubated at 20°C for 7 days, and mycelial morphology and subcellular structure were observed using SEM and TEM.

Mycelial growth was collected, washed three times with sterilized distilled water, transferred to MM–N liquid medium containing 4 mM phenylmethylsulfonyl fluoride, and incubated at 25°C for 4 h on a 200-rpm shaker. Fungal mass was then collected and fixed in 2.5% glutaraldehyde in 0.1 M phosphate buffer (pH 7.2) at 4°C overnight. The samples were then rinsed three times with phosphate buffer and fixed overnight at 4°C in 1% osmium tetroxide in 0.1 M cacodylate buffer (pH 7.0). After rinsing three times with phosphate buffer, the samples were dehydrated in an ethanol series, infiltrated with a graded series of epoxy resin in epoxy propane, and embedded in Epon 812 resin. The ultrathin sections were then stained in 2% uranium acetate followed by lead citrate, and subsequently visualized under a transmission electron microscope (Hitachi, H-7650) operating at 80 kV.

## RNA isolation

For each treatment sample, total RNA was extracted from frozen mycelium using Trizol reagent (Takara Bio, Japan) according to the manufacturer's instructions. RNase-free DNase I (Takara Bio, Japan) was used to digest traces of genomic DNA. RNA concentration was spectrophotometrically determined using the NanoPhotometer® spectrophotometer (IMPLEN, CA, USA), and RNA integrity and size distribution were assessed using 1% agarose gel electrophoresis.

## Transcriptomic analysis

cDNA library construction and Illumina sequencing were completed in Novogene Bioinfomatics Technology Company (Beijing, China) following a default Illumina stranded RNA protocol. Briefly, A total amount of 3 μg RNA per sample was used as input material for the RNA sample preparations. Sequencing libraries were generated using NEBNext® Ultra™ RNA Library Prep Kit for Illumina® (NEB, USA) following manufacturer's recommendations and index codes were added to attribute sequences to each sample. The clustering of the index-coded samples was performed on a cBot Cluster Generation System using TruSeq PE Cluster Kit v3-cBot-HS (Illumia) according to the manufacturer's instructions. After cluster generation, the library preparations were sequenced on an Illumina Hiseq platform and 150 bp paired-end reads were generated. We used three biological replicates of each sample. Differential expression analysis was performed using the DESeq R package (1.18.0) [51]. DESeq provide statistical routines for determining differential expression in digital gene expression data using a model based on the negative binomial distribution. The resulting P-values were adjusted using the Benjamini and Hochberg's approach for controlling the false discovery rate. Genes with an adjusted P-value <0.05 found by DESeq were assigned as differentially expressed. The differentially expressed genes (DEGs) were annotated using KEGG (Kyoto Encyclopedia of Genes and Genomes) database (http://www.genome.jp/kegg/). Gene Ontology (GO) enrichment analysis of differentially expressed genes was implemented by the GOseq R package, in which gene length bias was corrected. All of the raw reads are archived at the NCBI Sequence Read Archive (SRA) database (accession number: SRP212990).

## Quantitative real-time RT-PCR (RT-qPCR)

To confirm RNA-seq results, 20 DEGs were chosen for RT-qPCR analyses. According to manufacturer's instructions, RT-qPCR cycling conditions were as follows: initial denaturation for 3 min at 95˚C, 45 cycles of 3-s denaturation at 95˚C, 30-s annealing at 60˚C, and 30-s elongation at 60˚C. We used SG Fast qPCR Master Mix (High Rox) (ABI). With a minimum of two biological replicates, three technical replicates were subjected to RT-qPCR for each sample. Primers used in this study for RT-qPCR analyses are listed in S2 Table.

## Supporting information

**S1 Table. Gene expression changes in *Botrytis cinerea* cultivated with 100 μg/mL and without wuyiencin.**
(XLSX)

**S2 Table. Gene expression changes in *Botrytis cinerea* cultivated with 100 μg/mL and without (ck) wuyiencin, and validation of results by RT-qPCR.**
(XLSX)

**S3 Table. The information of metabolic pathways of the differentially expressed genes (DEGs) in *Botrytis cinerea* cultivated with 100 μg/mL and without wuyiencin.**
(XLS)

## Acknowledgments

We thank the native English-speaking scientists at Elixigen Company (Huntington Beach, California) for editing our manuscript.

## Author Contributions

**Conceptualization:** Kecheng Zhang.

**Data curation:** Qiuhe Wei, Beibei Ge.

**Formal analysis:** Beibei Ge.

**Funding acquisition:** Kecheng Zhang.

**Investigation:** Liming Shi, Kecheng Zhang.

**Methodology:** Liming Shi, Binghua Liu, Qiuhe Wei.

**Supervision:** Beibei Ge, Kecheng Zhang.

**Writing – original draft:** Liming Shi, Beibei Ge, Kecheng Zhang.

**Writing – review & editing:** Liming Shi, Kecheng Zhang.

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
