## [Decision Letter · Decision Letter 0]

27 Nov 2019

PONE-D-19-28852

Genome-wide Transcriptomic Analysis of the Response of Botrytis cinerea to Wuyiencin

PLOS ONE

Dear ProProfessor Zhang,

Thank you for submitting your manuscript to PLOS ONE. After careful consideration, we feel that it has merit but does not fully meet PLOS ONE’s publication criteria as it currently stands. Therefore, we invite you to submit a revised version of the manuscript that addresses the points raised during the review process.

We would appreciate receiving your revised manuscript by Jan 11 2020 11:59PM. To enhance the reproducibility of your results, we recommend that if applicable you deposit your laboratory protocols in protocols.io, where a protocol can be assigned its own identifier (DOI) such that it can be cited independently in the future. For instructions see: http://journals.plos.org/plosone/s/submission-guidelines#loc-laboratory-protocols

We look forward to receiving your revised manuscript.

Kind regards,

Kandasamy Ulaganathan

Academic Editor

PLOS ONE

Journal Requirements:

1. Thank you for including the following funding statement within your acknowledgements section of your manuscript; "This work was supported by the National Natural Science Foundation (31601684), the Special Fund for Basic Scientific Research of the Chinese Academy of Agricultural Sciences (Y2017JC12), and the National Key Research and Development Plan (2017YFD0201100). "

"NO"

2. We note that you are reporting an analysis of a microarray, next-generation sequencing, or deep sequencing data set. PLOS requires that authors comply with field-specific standards for preparation, recording, and deposition of data in repositories appropriate to their field. Please upload these data to a stable, public repository (such as ArrayExpress, Gene Expression Omnibus (GEO), DNA Data Bank of Japan (DDBJ), NCBI GenBank, NCBI Sequence Read Archive, or EMBL Nucleotide Sequence Database (ENA)). In your revised cover letter, please provide the relevant accession numbers that may be used to access these data. For a full list of recommended repositories, see http://journals.plos.org/plosone/s/data-availability#loc-omics or http://journals.plos.org/plosone/s/data-availability#loc-sequencing.

Additional Editor Comments (if provided):

You are requested to undertake a major revision of the paper addressing all the points raised by the reviewers. Please provide your reply for each of the comments made by the reviewers

Reviewers' comments:

Reviewer's Responses to Questions

**Comments to the Author**

1. Is the manuscript technically sound, and do the data support the conclusions?

Reviewer #1: Yes

Reviewer #2: Partly

Reviewer #3: Partly

Reviewer #4: Partly

Reviewer #5: No

2. Has the statistical analysis been performed appropriately and rigorously? 

Reviewer #1: Yes

Reviewer #2: Yes

Reviewer #3: Yes

Reviewer #4: No

Reviewer #5: No

3. Have the authors made all data underlying the findings in their manuscript fully available?

Reviewer #1: No

Reviewer #2: Yes

Reviewer #3: Yes

Reviewer #4: Yes

Reviewer #5: No

4. Is the manuscript presented in an intelligible fashion and written in standard English?

Reviewer #1: Yes

Reviewer #2: No

Reviewer #3: Yes

Reviewer #4: No

Reviewer #5: Yes

5. Review Comments to the Author

Reviewer #1: This is a decent study of genome-wide transcriptomic analysis of B. cinerea treated with wuyiencin. The authors identified 2067 differentially expressed genes, and performed and discussed pathway analysis, which revealed the molecular mechanisms of wuyiencin in treating B. cinerea.

Here are some minor questions.

1, Table S1 and Table S2 may be mis-labeled or uploaded in the opposite order. Can you double check?

2, Figure 2 looks nice. However, for the inner circle, I can barely read their colors and match them to the legend. Is there anyway to improve? Also for the third circle, what does the length of each bar represent?

Reviewer #2: The authors describe the inhibitory effects on Botrytis cinerea of the antifungal metabolite wuyiencin from the biological control bacterium Streptomyces ahygroscopicus var. wuyiensis, and present transcriptome data showing changes in gene expression in wuyiencin-treated B. cinerea compared to untreated controls.

The morphological effects of wuyiencin treatment are quite superficially, from the pictures shown in Fig. 1, it is difficult to draw any deeper conclusion about the molecular processes that are affected. The main part of the manuscript deals with the analyis of differential gene expression based on RNA sequencing. As expected from a potent growth inhibitor, many genes encoding metabolic enzymes are up- or downregulated, however, it is difficult to draw more specific conclusions about the molecular target(s) of wuyiencin. This applies also to the differential upregulation by wuyiencin of major plant cell degrading enzymes encoding genes (Bcxyn11A, Bcpg1/2, Bcpme1/2), which are interesting observations but don’t provide clear evidences towards a certain mode of action. In general, the interpretation of differential transcriptome studies taken at only one time point is very difficult because they are unable to describe the dynamics of gene expression changes induced by inhibitor treatment, and therefore yield only limited informations.

Part of the text is not written very well and carefully. Line numbers are missing. Some recent papers (e.g. about the latest B. cinerea genome sequence, van Kan et al., Mol Plant Pathol. 2017 18:75-89) have not been cited.

Reviewer #3: In this manuscript, Shi and colleagues present the transcriptome-wide response of Botrytis cinerea to exposure to the anti-fungal agent, wuyiencin. The manuscript is well written and provides an in-depth characterization of the genes which are differentially expressed in response to the drug. I had no major criticisms with the science, although I did find some of the results/conclusions a bit perplexing (discussed below) and felt that inclusion of p-values at certain points in the text would be informative to the reader (also discussed below). Aside from these two points, which can be easily fixed by some minor additions to the text, I feel that this manuscript is appropriate for publication in PLOS One.

Comments:

1. When first using the abbreviation for control (I assume this is what ck means) please define it (first paragraph of results).

2. When using phrases such as "almost significantly" please include p-values. My interpretation of "almost" may different than another reader's, particularly since the adjusted p-value cutoff was 0.05.

3. I found the upregulation of certain genes involved in pathogenesis, such as the CWDEs and the CAZymes, in response to the antifungal agent to be counter-intuitive. Perhaps a little more discussion on why this might be occurring would be helpful.

4. It is interesting that no genes involved in cell death are upregulated. In addition, the genes associated with stress responses that were identified in this study were all down-regulated.

5. Out of the pathways shown to be differentially expressed, could the authors point to a particular one which they believe is most likely the target of wuyiencin? The authors mention in the discussion that this work highlights potential mechanisms but I think I missed the ones that they predict to be most likely.

Reviewer #4: In order to elucidate the effects of wuyiencin on the transcriptional regulation in B. cinerea, the authors used genome-wide transcriptomic analysis of B. cinerea treated with wuyiencin.

The found that putative genes involved in amino acid metabolism were influenced in response to wuyiencin treatment. Moreover, the suggest expression of genes involved in protein synthesis and energy metabolism and of those encoding ATP-binding cassette transporters was markedly upregulated, whereas that of genes participating in DNA replication, cell cycle, and stress response was downregulated. Furthermore, wuyiencin resulted in mycelial malformation and negatively influenced cell growth rate and conidial yield in B. cinerea. Even though the subject is very important and the transcriptional analysis can give hint on wuyencin mode of actions this paper is mainly descriptive and not supported by functional analysis.

Major concerns:

1. Is there any correlation between the phenotypic influence by SEM and TEM to the transcriptional changes- this need to be addresses and discussed.

2. Lack of functional analysis of the effect demonstrated by transcriptional changes (e.g. are AA levels or metabolism really changed? ; are CWDE active?; Do cell cycle/DNA replication impaired?; how can athoures conclude that wuyiencin active efflux is occurs?)

3. Not enough details on the RNA-seq experiment and analysis such as PCA , reads number etc.

4. Missing statistic on qRT-PCR analysis

5. Figure 2 is unreadable! - must be replaced with better way of demonstration the transcriptional changes by GO- annotation diagrams or pies., KEGG analysis is missing and need to be added

6. The authors fail to discuss how come some of the pathway they suggested to be involved is upregulate but other genes in the same pathway are down regulated- in AA metabolism or CWDE (Figures 3 and 4)?

7. Is ROS effected by wuyiencin, may check similar effect demonstrated by Vogel et al., 2011

8. Authors fail to discuss their results and explain properly how specific the changes wuyiencin are and not part of general stress response??

9. Reference 11 is very important backbone for this paper and it is not available in pubmed

Minor concerns:

1. Figures legends are not sufficient

2. Statistic is missing Figures 3-4

3. Fig1. no size bars, arrows are not defined, pigment is not shown?

4. wuyiencin is not biological control agent??

5. Is there any correlation between RNAseq and qRT-PCR?

6. Sometime there is no correlation between genes marked on Fig 3 to the text in result section

7. In results section may parts belong to discussion

8. Conidial germination S1 figure should be in main text, also demonstrate other results on conidia

9. Why control called ck?? It is very confusing

10. Some earlier studies references are missing (e.g. in results second part line 3)

11. In discussion part paragraph 4 “ In the past…” is not connected and need to be removed

12. The manuscript is not written fluently and very hard to read

13. The MS must go through English professional editing

Reviewer #5: The manuscript titled "Genome-wide Transcriptomic Analysis of the Response of Botrytis cinerea to Wuyiencin" by Shi et al is well written and provides a solid base for the future investigation of the mode of action of Wuyiencin however, the authors did not provide access to their data nor did they provide citations for how their analyses were carried out. Just stating that a given software package was used does provide sufficient information to asses the statistical analyses carried out. A further point is that the quality of the images in the figure in the version of the manuscript I dowloaded was insufficient to illustrate what the authors indicated in the text especially for figure 1 and 2. A final major point is that there are instances in where the RT-qPCR data does not fully support the RNA-seq data and as such the athors need to moderate statments that suggest it is fully consistant with the RNA-seq data

Apart from these major issues that need to be addressed the authors also need to improve the language associated with their analyse of the data.

For example In the abstract the sentence beginning "We could identify 2067 differentially .." should be edited to improve clarity

Change the phrase "remarkably influence" to improve precision of the statement give specifics, up regulated ? down regulated? or even change the proceeding structure to something like " transcript levels of genes involved in amino acid metabolism and those encoding putative secreted proteins were altered..."

IN this regard it is more precise that the authors refer to altered transcript levels since they are not specifically measuring gene expression.

Also the last sentence of the abstract is too general the data presented does not suppot this statement so it should be edited to clarify what the data does support.

In the first paragraph of the introduction edit the phrase "relatively suitable condition" to more precisely indicate which conditions are required for conidiation.

Introduction second paragraph the evidence is not conclusive enough to make the statment that differential expression of enzymes is the only basis for B. cinerea infection of multiple hosts.

QRT_PCR should be RT-qPCR since it is the PCR that is quantitative and RT-qPCR does not validate RNA-seq data it provides data that support that obtained by RNA-seq.

IN the results, Figure 2 does not illustrate clearly the number of genes with altered transcript levels

KEGG does not simply classify genes it assess the representation of specific functional groups of genes edit the text to clarify this indicating that the DEGs were enriched in the following functional categories ...

The phrase almost significantly down regulated is meaningless indicate specifically what the data shows.

Statements like "Protein synthesis, although an exceedingly complicated process ..." are overly obvious and add nothing to the text add such statements to indicate more precisely what the authors are trying to convey

The sentence "These findings suggest that B. cinerea uses different protein synthesis metabolism when exposed to phytoalexin from plants and antibiotic in microorganisms." is not clear do the authors mean " ... and when exposed to antibiotics from microorganisms"?

The phrase "most significant" is not meaningful, edit to clarify precisely what the authors intend to convey

The statement "The downregulation of the expression of these genes is directly related to cell growth

rate impairment in B. cinerea upon wuyiencin treatmen." is poorly worded and not support. The authors need to cite data that supports the contention that altering the transcript levels of these genes alters growth rate

To further validate was not the reason RT-qPCR was carried out it was carried out to assess if RNA_seq data was supportable by a different method of analysis

The authors state "significantly downregulated" but never define this nor do they provide sufficient information on their statistical analyses such that this can be independently assessed. this must be edited for clarification.

Sun et al is referenced but no date is provided for this reference

The statement "We strongly believe" has no place in a scientific publication. This is not religion, evidence must be presented to support conclusions.

The authors state "...in this study, we identified candidate pathways and target genes that may offer insights into the protective and antagonistic mechanisms underlying the action of biological control agents." but do not eleborate on what information they can gain from the data about the potential mode of action of the antifungal used and how that relates to other antifungal of the same class

In the conclusion the phrase "actual situations" is meaningless the authors must edit this phrase and the sentence it is in to provide clarity of the point they are trying to convey

The authors must define how they denatured the RNA during agarose gel electrophoresis

The authors must provide more information on how they used DESeq R and provide a citation for this analysis

Eliminate any reference of RT-qPCR being carried out to confirm RN-seq data this suggests that you had expectations of what the data would show which would have influenced interpretation of he reults

6. PLOS authors have the option to publish the peer review history of their article (what does this mean?). If published, this will include your full peer review and any attached files.

Reviewer #1: No

Reviewer #2: Yes: Matthias Hahn

Reviewer #3: Yes: Andrew D. L. Nelson

Reviewer #4: No

Reviewer #5: No

---

## [Author Response · Author response to Decision Letter 0]

11 Feb 2020

Reviewers' comments:

Reviewer's Responses to Questions

Comments to the Author

1. Is the manuscript technically sound, and do the data support the conclusions?

Reviewer #1: Yes

Reviewer #2: Partly

Reviewer #3: Partly

Reviewer #4: Partly

Reviewer #5: No

2. Has the statistical analysis been performed appropriately and rigorously? 

Reviewer #1: Yes

Reviewer #2: Yes

Reviewer #3: Yes

Reviewer #4: No

Reviewer #5: No

3. Have the authors made all data underlying the findings in their manuscript fully available?

Reviewer #1: No

Reviewer #2: Yes

Reviewer #3: Yes

Reviewer #4: Yes

Reviewer #5: No

4. Is the manuscript presented in an intelligible fashion and written in standard English?

Reviewer #1: Yes

Reviewer #2: No

Reviewer #3: Yes

Reviewer #4: No

Reviewer #5: Yes

Review Comments to the Author

Reviewer #1: This is a decent study of genome-wide transcriptomic analysis of B. cinerea treated with wuyiencin. The authors identified 2067 differentially expressed genes, and performed and discussed pathway analysis, which revealed the molecular mechanisms of wuyiencin in treating B. cinerea.

Here are some minor questions.

1, Table S1 and Table S2 may be mis-labeled or uploaded in the opposite order. Can you double check?

Answer: Table S1 and Table S2 were checked and revised.

2, Figure 2 looks nice. However, for the inner circle, I can barely read their colors and match them to the legend. Is there anyway to improve? Also for the third circle, what does the length of each bar represent? 

Answer: Figure 2 was revised. The legend of Figure 2 has been modified. Thanks for your kind comment, we put the information of metabolic pathways of DEGs in the supplemental Table S3.

For instance:

Figure 2. Circular map of B05.10 genome and genes that were differentially expressed (DEGs) upon wuyiencin treatment. Moving inward, the outer two rings show the length (red ring) and gene (green ring) density of every chromosome, respectively. The third ring represents differentially expressed genes (DEGs): upregulated and downregulated DEGs are highlighted as red and blue bars, respectively. The length of each bar represent the fold change. 

Reviewer #2: The authors describe the inhibitory effects on Botrytis cinerea of the antifungal metabolite wuyiencin from the biological control bacterium Streptomyces ahygroscopicus var. wuyiensis, and present transcriptome data showing changes in gene expression in wuyiencin-treated B. cinerea compared to untreated controls.

The morphological effects of wuyiencin treatment are quite superficially, from the pictures shown in Fig. 1, it is difficult to draw any deeper conclusion about the molecular processes that are affected. The main part of the manuscript deals with the analyis of differential gene expression based on RNA sequencing. As expected from a potent growth inhibitor, many genes encoding metabolic enzymes are up- or downregulated, however, it is difficult to draw more specific conclusions about the molecular target(s) of wuyiencin. This applies also to the differential upregulation by wuyiencin of major plant cell degrading enzymes encoding genes (Bcxyn11A, Bcpg1/2, Bcpme1/2), which are interesting observations but don’t provide clear evidences towards a certain mode of action. In general, the interpretation of differential transcriptome studies taken at only one time point is very difficult because they are unable to describe the dynamics of gene expression changes induced by inhibitor treatment, and therefore yield only limited informations.

Part of the text is not written very well and carefully. Line numbers are missing. Some recent papers (e.g. about the latest B. cinerea genome sequence, van Kan et al., Mol Plant Pathol. 2017 18:75-89) have not been cited.

Answer: Line numbers are adding and the recent paper have been cited. 

Reviewer #3: In this manuscript, Shi and colleagues present the transcriptome-wide response of Botrytis cinerea to exposure to the anti-fungal agent, wuyiencin. The manuscript is well written and provides an in-depth characterization of the genes which are differentially expressed in response to the drug. I had no major criticisms with the science, although I did find some of the results/conclusions a bit perplexing (discussed below) and felt that inclusion of p-values at certain points in the text would be informative to the reader (also discussed below). Aside from these two points, which can be easily fixed by some minor additions to the text, I feel that this manuscript is appropriate for publication in PLOS One.

Comments:

1. When first using the abbreviation for control (I assume this is what ck means) please define it (first paragraph of results). 

Answer: In the first paragraph of results (Line 104), it was revised.

2. When using phrases such as "almost significantly" please include p-values. My interpretation of "almost" may different than another reader's, particularly since the adjusted p-value cutoff was 0.05.

Answer: Line 151, "almost" was deleted, to avoid misunderstanding. 

3. I found the upregulation of certain genes involved in pathogenesis, such as the CWDEs and the CAZymes, in response to the antifungal agent to be counter-intuitive. Perhaps a little more discussion on why this might be occurring would be helpful.

Answer: The related content was in Discussion part. 

4. It is interesting that no genes involved in cell death are upregulated. In addition, the genes associated with stress responses that were identified in this study were all down-regulated.

Answer: Yes.

5. Out of the pathways shown to be differentially expressed, could the authors point to a particular one which they believe is most likely the target of wuyiencin? The authors mention in the discussion that this work highlights potential mechanisms but I think I missed the ones that they predict to be most likely.

Answer: The most likely target of wuyiencin is the cell growth and differentiation of Botrytis cinerea, especially in amino acid metabolism, protein synthesis and energy metabolism. 

Reviewer #4: In order to elucidate the effects of wuyiencin on the transcriptional regulation in B. cinerea, the authors used genome-wide transcriptomic analysis of B. cinerea treated with wuyiencin.

The found that putative genes involved in amino acid metabolism were influenced in response to wuyiencin treatment. Moreover, the suggest expression of genes involved in protein synthesis and energy metabolism and of those encoding ATP-binding cassette transporters was markedly upregulated, whereas that of genes participating in DNA replication, cell cycle, and stress response was downregulated. Furthermore, wuyiencin resulted in mycelial malformation and negatively influenced cell growth rate and conidial yield in B. cinerea. Even though the subject is very important and the transcriptional analysis can give hint on wuyencin mode of actions this paper is mainly descriptive and not supported by functional analysis.

Major concerns:

1. Is there any correlation between the phenotypic influence by SEM and TEM to the transcriptional changes- this need to be addresses and discussed.

Answer: In this study, we didn’t find any relationship between SEM and TEM to the transcriptional changes-. For the further study, the phenotypic change of unique gene or pathway may easily to observed by SEM and TEM.

2. Lack of functional analysis of the effect demonstrated by transcriptional changes (e.g. are AA levels or metabolism really changed? ; are CWDE active?; Do cell cycle/DNA replication impaired?; how can athoures conclude that wuyiencin active efflux is occurs?)

Answer: According to the analysis of transcriptome, the genes relating to amino acid metabolism, protein synthesis and cell cycle/DNA replication are differentially expressed. And the expression of some genes in these metabolism are verified by RT-qPCR.

3. Not enough details on the RNA-seq experiment and analysis such as PCA , reads number etc.

Answer: The details on RNA-seq experiment and analysis were added in Methods part.

4. Missing statistic on qRT-PCR analysis

Answer: The statistic result of qRT-PCR analysis was added in Table S2.

5. Figure 2 is unreadable! - must be replaced with better way of demonstration the transcriptional changes by GO- annotation diagrams or pies., KEGG analysis is missing and need to be added.

Answer: Figure 2 was revised and KEGG analysis is added in Method part.

6. The authors fail to discuss how come some of the pathway they suggested to be involved is upregulate but other genes in the same pathway are down regulated- in AA metabolism or CWDE (Figures 3 and 4)?

Answer: This is difficult to explain the reversed gene expression just from transcriptional data. The affection of wuyiencin may trigger the host gene feedback regulation for survival. This speculate need more single-gene experimental data to support.

7. Is ROS effected by wuyiencin, may check similar effect demonstrated by Vogel et al., 2011

Answer: We didn’t find this ref, but ROS effected by wuyiencin was similar with part of general stress response, and this was further discussed in revised manuscript in the second paragraph from bottom of discussion.

8. Authors fail to discuss their results and explain properly how specific the changes wuyiencin are and not part of general stress response??

Answer: From this study, we just know that there are some similar changes that caused by the wuyiencin compared with general stress response. The further discussion was added to the revised manuscript in the second paragraph from bottom of discussion and add the related refs below.

Gessler, N. N.; Aver'yanov, A. A.; Belozerskaya, T. A. Reactive oxygen species in regulation of fungal development. Biochemistry (Mosc).2007, 72, 1091-109.

Heller, J.; Tudzynski, P. Reactive oxygen species in phytopathogenic fungi: signaling, development, and disease. Annu Rev Phytopathol.2011, 49, 369-90.

9. Reference 11 is very important backbone for this paper and it is not available in pubmed

Answer: The reference 11can be found in Baiduxueshu website (www. xueshu.baidu.com/).

Minor concerns:

1. Figures legends are not sufficient

Answer: Figures legends had been revised. 

2. Statistic is missing Figures 3-4 

Answer: We used the unique mapped reads to annotate the readcount of each genes. And the significant difference analysis about the expression level of each gene was carried out by DESeq. And there is just one readcount number for each gene in the result file. Statistic is contained in the calculating of P-values.

3. Fig1. no size bars, arrows are not defined, pigment is not shown?

Answer: Size bars were added. Pigment was calculated using blood counting chamber and showed in histogram of Fig 1E.

4. wuyiencin is not biological control agent??

Answer: Wuyiencin is biological control agent.

5. Is there any correlation between RNAseq and qRT-PCR?

Answer: The selected genes used in qRT-PCR are all representative ones which significant differentially expressed in RNAseq. The qRT-PCR is the verification of RNAseq result and they are consistent.

6. Sometime there is no correlation between genes marked on Fig 3 to the text in result section

Answer: The key genes in Fig 3 are mainly involved in amino acid metabolism. And some of them also participate in protein synthesis or carbon and energy metabolism. Some sentences have been adjusted in order to the correlation between genes marked on Fig 3 to the text.

7. In results section may parts belong to discussion.

Answer: Yes.

8. Conidial germination S1 figure should be in main text, also demonstrate other results on conidia

Answer: Figure S1 was added to Figure 1 as 1E.

9. Why control called ck?? It is very confusing

Answer: In the first paragraph of results (Line 104), it was revised.

10. Some earlier studies references are missing (e.g. in results second part line 3)

Answer: The reference had been added. 

11. In discussion part paragraph 4 “ In the past…” is not connected and need to be removed

Answer: It was revised.

12. The manuscript is not written fluently and very hard to read.

Answer: We tried our best to improve the manuscript and made some changes for better reading.

13. The MS must go through English professional editing.

Answer: Yes.

Reviewer #5: The manuscript titled "Genome-wide Transcriptomic Analysis of the Response of Botrytis cinerea to Wuyiencin" by Shi et al is well written and provides a solid base for the future investigation of the mode of action of Wuyiencin 

however, the authors did not provide access to their data nor did they provide citations for how their analyses were carried out. Just stating that a given software package was used does provide sufficient information to assess the statistical analyses carried out. 

Answer: All of the raw reads have been uploaded to NCBI Sequence Read Archive (SRA) database. And more detailed analysis method that we carried out and the accession number has been added to Methods part and data availability statement.

A further point is that the quality of the images in the figure in the version of the manuscript I dowloaded was insufficient to illustrate what the authors indicated in the text especially for figure 1 and 2. 

Answer: The quality of the images were improved.

A final major point is that there are instances in where the RT-qPCR data does not fully support the RNA-seq data and as such the athors need to moderate statments that suggest it is fully consistant with the RNA-seq data

Answer: The two results are not fully consistant, but the overall trend is the same. The sentence has been revised to express more accurately.

Apart from these major issues that need to be addressed the authors also need to improve the language associated with their analyse of the data.

For example In the abstract the sentence beginning "We could identify 2067 differentially .." should be edited to improve clarity. Change the phrase "remarkably influence" to improve precision of the statement give specifics, up regulated ? down regulated? or even change the proceeding structure to something like " transcript levels of genes involved in amino acid metabolism and those encoding putative secreted proteins were altered..."IN this regard it is more precise that the authors refer to altered transcript levels since they are not specifically measuring gene expression. 

Answer: The sentences in the abstract and other part had been revised to improve clarity.

Also the last sentence of the abstract is too general the data presented does not suppot this statement so it should be edited to clarify what the data does support.

Answer: The last sentence of the abstract has been modified according to reviewer’s suggestion.

In the first paragraph of the introduction edit the phrase "relatively suitable condition" to more precisely indicate which conditions are required for conidiation.

Answer: The sentences in the first paragraph of the introduction had been revised to improve clarity.

Introduction second paragraph the evidence is not conclusive enough to make the statment that differential expression of enzymes is the only basis for B. cinerea infection of multiple hosts.

Answer: The sentences in the second paragraph of the introduction had been revised to improve clarity, and the differential expression of enzymes is just one way for B. cinerea infection of multiple hosts.

QRT_PCR should be RT-qPCR since it is the PCR that is quantitative and RT-qPCR does not validate RNA-seq data it provides data that support that obtained by RNA-seq.

Answer: The qRT-PCR statement and related sentences had been corrected.

IN the results, Figure 2 does not illustrate clearly the number of genes with altered transcript levels

Answer: Figure 2 was revised.

KEGG does not simply classify genes it assess the representation of specific functional groups of genes edit the text to clarify this indicating that the DEGs were enriched in the following functional categories ...

Answer: Yes.

The phrase almost significantly down regulated is meaningless indicate specifically what the data shows.

Answer: The sentences in line 151 had been revised to improve clarity.

Statements like "Protein synthesis, although an exceedingly complicated process ..." are overly obvious and add nothing to the text add such statements to indicate more precisely what the authors are trying to convey

Answer: The sentences in line 191 had been deleted.

The sentence "These findings suggest that B. cinerea uses different protein synthesis metabolism when exposed to phytoalexin from plants and antibiotic in microorganisms." is not clear do the authors mean " ... and when exposed to antibiotics from microorganisms"?

Answer: The sentences in line 213 had been revised to improve clarity.

The phrase "most significant" is not meaningful, edit to clarify precisely what the authors intend to convey

Answer: The sentences in line 216 had been revised to improve clarity.

The statement "The downregulation of the expression of these genes is directly related to cell growth rate impairment in B. cinerea upon wuyiencin treatmen." is poorly worded and not support. The authors need to cite data that supports the contention that altering the transcript levels of these genes alters growth rate

Answer: The sentences in line 325 had been revised to improve clarity.

To further validate was not the reason RT-qPCR was carried out it was carried out to assess if RNA_seq data was supportable by a different method of analysis

Answer: The sentences in line 326 had been revised to improve clarity.

The authors state "significantly downregulated" but never define this nor do they provide sufficient information on their statistical analyses such that this can be independently assessed. this must be edited for clarification.

Answer: The sentences in line342 and line345 were added ‘(Table S1)’ which could provide information.

Sun et al is referenced but no date is provided for this reference

Answer: The reference were added here (Line 381). 

The statement "We strongly believe" has no place in a scientific publication. This is not religion, evidence must be presented to support conclusions.

Answer: The sentences in line 435 had been revised to improve clarity.

The authors state "...in this study, we identified candidate pathways and target genes that may offer insights into the protective and antagonistic mechanisms underlying the action of biological control agents." but do not eleborate on what information they can gain from the data about the potential mode of action of the antifungal used and how that relates to other antifungal of the same class

Answer: According to the analysis of transcriptome, the genes relating to amino acid metabolism, protein synthesis and cell cycle/DNA replication are differentially expressed. And the expression of some genes in these metabolism are verified by RT-qPCR. The most likely target of wuyiencin is the cell growth and differentiation of Botrytis cinerea, especially in amino acid metabolism, protein synthesis and energy metabolism.

In the conclusion the phrase "actual situations" is meaningless the authors must edit this phrase and the sentence it is in to provide clarity of the point they are trying to convey

Answer: The sentences in line 441 had been revised to improve clarity and the phrase "actual situations" means the cultivation environment in actual field.

The authors must define how they denatured the RNA during agarose gel electrophoresis

Answer: We did not denature the RNA during agarose gel electrophoresis. Only RNA integrity and size distribution were assessed using 1% agarose gel electrophoresis.

The authors must provide more information on how they used DESeq R and provide a citation for this analysis

Answer: The Methods part Transcriptomic analysis were supplemented and the reference had been cited.

Eliminate any reference of RT-qPCR being carried out to confirm RNA-seq data this suggests that you had expectations of what the data would show which would have influenced interpretation of the results

Answer: To confirm RNA-seq results, 20 DEGs were chosen for RT-qPCR randomly without expectations.

---

## [Decision Letter · Decision Letter 1]

27 Mar 2020

Genome-wide Transcriptomic Analysis of the Response of Botrytis cinerea to Wuyiencin

PONE-D-19-28852R1

Dear Dr. Zhang,

We are pleased to inform you that your manuscript has been judged scientifically suitable for publication and will be formally accepted for publication once it complies with all outstanding technical requirements.

With kind regards,

Kandasamy Ulaganathan

Academic Editor

PLOS ONE

Additional Editor Comments (optional):

Reviewers' comments:

Reviewer's Responses to Questions

**Comments to the Author**

1. If the authors have adequately addressed your comments raised in a previous round of review and you feel that this manuscript is now acceptable for publication, you may indicate that here to bypass the “Comments to the Author” section, enter your conflict of interest statement in the “Confidential to Editor” section, and submit your "Accept" recommendation.

Reviewer #1: All comments have been addressed

Reviewer #3: (No Response)

Reviewer #5: All comments have been addressed

2. Is the manuscript technically sound, and do the data support the conclusions?

Reviewer #1: Yes

Reviewer #3: Yes

Reviewer #5: Yes

3. Has the statistical analysis been performed appropriately and rigorously? 

Reviewer #1: Yes

Reviewer #3: Yes

Reviewer #5: Yes

4. Have the authors made all data underlying the findings in their manuscript fully available?

Reviewer #1: Yes

Reviewer #3: Yes

Reviewer #5: Yes

5. Is the manuscript presented in an intelligible fashion and written in standard English?

Reviewer #1: Yes

Reviewer #3: Yes

Reviewer #5: Yes

6. Review Comments to the Author

Reviewer #1: The author has addressed all my concerns. I have no further questions. I would suggest to accept this paper.

Reviewer #3: I thank the authors for addressing my concerns. I still do not think that they really address in the discussion that they observe downregulation of stress response genes (lines 420-423) and upregulation of genes that a naive reviewer would assume to help the pathogen be more virulent (lines 277-283) or better degrade the cell wall of their host. Finally, the authors mention on line 323-324 that downregulation of the expression of these genes (Bcin09g01360, etc) are directly related to cell growth impairment in B. cinerea... This sentence is speculation and as such should be moved to the discussion or changed to "may be related".

Overall though, I think the data will be useful to the community. Aside from the above points and a few areas where grammar needs to be addressed, I think this manuscript is suitable for publication.

Reviewer #5: I feel that the authors have addressed the concerns I raised adequately. The improved images make a big difference. There is still some minor issues with wording but it is acceptable.

7. PLOS authors have the option to publish the peer review history of their article (what does this mean?). If published, this will include your full peer review and any attached files.

Reviewer #1: No

Reviewer #3: Yes: Andrew Nelson

Reviewer #5: No

---

## [Editor Report · Acceptance letter]

20 Apr 2020

PONE-D-19-28852R1 

Genome-wide Transcriptomic Analysis of the Response of *Botrytis cinerea* to Wuyiencin 

Dear Dr. Zhang:

I am pleased to inform you that your manuscript has been deemed suitable for publication in PLOS ONE. Congratulations! Your manuscript is now with our production department. 

With kind regards,

on behalf of

Dr. Kandasamy Ulaganathan 

Academic Editor

PLOS ONE